# β-Catenin Activation in Hepatocellular Cancer: Implications in Biology and Therapy

**DOI:** 10.3390/cancers13081830

**Published:** 2021-04-12

**Authors:** Yekaterina Krutsenko, Aatur D. Singhi, Satdarshan P. Monga

**Affiliations:** Department of Pathology and Pittsburgh Liver Research Center, University of Pittsburgh and University of Pittsburgh Medical Center, Pittsburgh, PA 15261, USA; yek14@pitt.edu (Y.K.); singhiad@upmc.edu (A.D.S.)

**Keywords:** β-catenin mutations, tumor metabolism, tumor immunology, molecular therapeutics, precision medicine

## Abstract

**Simple Summary:**

Liver cancer is a dreadful tumor which has gradually increased in incidence all around the world. One major driver of liver cancer is the Wnt–β-catenin pathway which is active in a subset of these tumors. While this pathway is normally important in liver development, regeneration and homeostasis, it’s excessive activation due to mutations, is detrimental and leads to tumor cell growth, making it an important therapeutic target. There are also some unique characteristics of this pathway activation in liver cancer. It makes the tumor addicted to specific amino acids and in turn to mTOR signaling, which can be treated by certain existing therapies. In addition, activation of the Wnt–β-catenin in liver cancer appears to alter the immune cell landscape making it less likely to respond to the new immuno-oncology treatments. Thus, Wnt–β-catenin active tumors may need to be treated differently than non-Wnt–β-catenin active tumors.

**Abstract:**

Hepatocellular cancer (HCC), the most common primary liver tumor, has been gradually growing in incidence globally. The whole-genome and whole-exome sequencing of HCC has led to an improved understanding of the molecular drivers of this tumor type. Activation of the Wnt signaling pathway, mostly due to stabilizing missense mutations in its downstream effector β-catenin (encoded by *CTNNB1*) or loss-of-function mutations in *AXIN1* (the gene which encodes for Axin-1, an essential protein for β-catenin degradation), are seen in a major subset of HCC. Because of the important role of β-catenin in liver pathobiology, its role in HCC has been extensively investigated. In fact, *CTNNB1* mutations have been shown to have a trunk role. β-Catenin has been shown to play an important role in regulating tumor cell proliferation and survival and in tumor angiogenesis, due to a host of target genes regulated by the β-catenin transactivation of its transcriptional factor TCF. Proof-of-concept preclinical studies have shown β-catenin to be a highly relevant therapeutic target in *CTNNB1*-mutated HCCs. More recently, studies have revealed a unique role of β-catenin activation in regulating both tumor metabolism as well as the tumor immune microenvironment. Both these roles have notable implications for the development of novel therapies for HCC. Thus, β-catenin has a pertinent role in driving HCC development and maintenance of this tumor-type, and could be a highly relevant therapeutic target in a subset of HCC cases.

## 1. The Wnt–β-Catenin Signaling Pathway

The protein later termed Wnt1 was first identified almost 40 years ago in the context of its proto-oncogenic nature [1,2]. Subsequent studies have characterized Wnt1 itself, as well as other highly conserved components of Wnt signaling, as a key mediator involved not only in tumorigenesis, but also in the fundamental cellular processes governing embryonic development and adult tissue homeostasis [3,4]. Yet, the vital role of aberrant Wnt signaling in cancer initiation and progression remains one of the most intriguing and vital themes in the field. The Wnt pathway involves a multitude of components, including ligands, receptors, and co-receptors acting in autocrine, paracrine, and endocrine fashion to regulate the processes of cell fate determination, proliferation, and polarity, among others [2,4,5]. Structural and functional classification has indicated the existence of several distinct Wnt signaling pathways, which can be broadly subdivided based on the involvement of β-catenin. β-Catenin-dependent canonical Wnt signaling remains arguably the most investigated branch.

In the canonical pathway, the control of the Wnt-dependent cellular processes is achieved by a tight regulation of the amount of β-catenin—a transcriptional co-activator and a regulator of cell–cell adhesion. Normally, in the absence of Wnt signals, cytosolic levels of β-catenin remain low due to continuous proteasomal degradation of the protein, initiated by its destruction complex. The complex, composed of the scaffold Axin, tumor-suppressor adenomatous polyposis coli (*APC*) gene product, and diversin, also includes two kinases, casein kinase 1 (CK1), and glycogen synthase kinase 3 (GSK3), which sequentially phosphorylate β-catenin, priming it for recognition by the ubiquitin ligase β-TrCP [1]. In the absence of negative regulation, the glycosylation and palmitoylation of Wnt glycoproteins allows their biological activity to in turn activate the Wnt–β-catenin signaling. The cascade is induced by the binding of secreted Wnts to the seven transmembrane G-protein-coupled Frizzled (Fz) receptors located at the plasma membrane [5]. The binding initiates the formation of a multicomponent complex consisting of Wnt ligand, Frizzled, and its co-receptor LRP (low-density lipoprotein receptor-related protein) 6 or 5 [6]. This, in turn, signals for the recruitment of Dishevelled (Dvl), and results in the phosphorylation of LRP5/6, thereby providing a docking site for the Axin and tethering it to the cell membrane, which eventually renders the β-catenin destruction complex inactive. Thus, the presence of Wnt ligands interferes with the sequestration of β-catenin and its subsequent ubiquitination, thereby stabilizing the protein in cytoplasm. This allows for the nuclear translocation of β-catenin, where it triggers the expression of Wnt-induced genes (i.e., Cyclin D1, c-Myc, vascular endothelial growth factor (VEGF), interleukin-8 (IL-8), etc.) by acting as transcriptional co-activator in conjunction with T-cell factor (TCF) and lymphoid enhancer factor (LEF) DNA-binding proteins [7].

## 2. Wnt–β-Catenin Signaling in Liver Pathophysiology

The central role of the canonical Wnt–β-catenin signaling pathway in multiple aspects of normal cell functioning and in pathobiological processes is especially eminent in liver [8,9,10,11]. There, β-catenin orchestrates embryonic development, patterning, adult tissue metabolism, proliferation, and regeneration. While discussing the many facets of β-catenin signaling as a component of the Wnt pathway is outside the scope of the current review, we would like to remind the readers of a few pertinent concepts that are also relevant in hepatocellular cancer (HCC).

### 2.1. Wnt–β-Catenin Signaling in Hepatic Development

β-Catenin was first reported to be active in normal mouse and chick embryonic liver development almost two decades ago [12,13,14]. β-Catenin was seen to be active in stages of hepatic development which showed proliferating hepatoblasts and immature hepatocytes. When mouse embryonic liver cultures were propagated in the presence of antisense oligonucleotides against the β-catenin gene, there was a notable deficit in the resident cell proliferation. This was later verified by conditional deletion of the β-catenin gene or via activation of β-catenin through APC gene loss from mouse hepatoblasts in vivo [15,16]. In addition to these observations, both in vitro and in vivo studies showed a dramatic compromise in hepatocyte maturation. This was seen as the maintenance of hepatoblast markers in the hepatocytes in the β-catenin absent or knocked-down livers, as well as by deficient markers of mature fetal hepatocytes, including glycogen [16]. Thus, β-catenin plays a role in both the proliferation of immature hepatocytes and hepatoblasts during earlier stages of hepatic development, but plays an equally important role in the maturation of immature hepatocytes during later stages. These temporal targets of β-catenin include c-myc and cyclin-D1 for proliferation, as well as CEBPα and as-yet unknown targets, which are likely distinct from its well-known zone-3 targets in adult liver [16]. It is also worth mentioning that, after birth, there is a postnatal growth spurt in livers from postnatal day 5 to about 25 days, after which the liver is mostly quiescent, showing minimal hepatocyte turnover [17]. β-Catenin signaling is also a major contributor of the postnatal wave of hepatocyte proliferation, and in its absence there is a decreased growth spurt which leaves liver-specific β-catenin knockout mice with around 15% lower liver-weight to body-weight ratio (LW/BW).

### 2.2. Wnt–β-Catenin Signaling in Liver Regeneration

Livers possess a unique feature of regeneration following surgical resection or toxicant-induced injury to regain its lost mass within days to weeks. The liver does so without any progenitor cell activation but via the replication of resident hepatocytes (and other cells) in the liver [18]. Wnt–β-Catenin signaling has been shown to be a key component of the normal molecular machinery of the liver following surgical resection [19]. Within hours of two-thirds hepatectomy, there is a nuclear translocation of β-catenin in hepatocytes and the appearance of β-catenin–TCF complex [20,21]. This is sustained for almost the first 48 h of regeneration. Using several genetic knockout mouse models, it appears that Wnt2 and Wnt9b are massively upregulated in hepatic sinusoidal endothelial cells and less so in monocytes/macrophages at 12 h after hepatectomy (earliest time point examined through individual cell-type isolation after surgery), followed by the engagement of Fzd-LRP5/6, resulting in the activation of β-catenin–TCF4 to regulate cyclin-D1 gene transcription [19]. The increased cyclin-D1 observed during this time allows for hepatocyte G1–S phase transition and eventually contributes to timely hepatocyte proliferation and the recovery of hepatic mass [22]. The absence of Wntless from endothelial cells (and less so macrophages) or the absence of LRP5 and 6 from hepatocytes or the absence of β-catenin from hepatocytes, all lead to a notable deficit in cyclin-D1 expression and a dramatically lower hepatocyte proliferation at 40–48 h after two-thirds hepatectomy [23,24,25,26,27]. Livers eventually recover in all models, despite a notable delay in restitution, and the mechanisms allowing for recovery in the absence of Wnt–β-catenin signaling remain unknown at this time. A similar role of the pathway during hepatocyte proliferation has also been reported after injury from acetaminophen, carbon-tetrachloride, diethoxycarbonyl dihydrocollidine, choline-deficient ethionine supplemented diet, and in Mdr2 knockout mice, making Wnt–β-catenin signaling a global hepatic repair pathway [28,29,30,31,32,33].

Intriguingly, a recent study also showed an important role of the Wnt–β-catenin pathway in serving a dual role of not only inducing hepatocyte proliferation but also maintaining hepatocyte function during liver regeneration after surgical resection, as well as after acetaminophen-induced injury and repair. Using single-cell RNA-sequencing, Walesky et al. showed a clever “division of labor” by the hepatocytes in the remnant liver following surgery or toxicant injury [34]. This strategy allows liver to maintain function even while it is proliferating, as distinct subsets of hepatocytes acquire proliferative versus hepatocyte-function phenotype, as shown by gene expression studies. Intriguingly, both these functions are regulated by the Wnt–β-catenin pathway; the cell source of the Wnt for regulating the hepatocyte function by β-catenin appears to be macrophages and not sinusoidal endothelial cells, which are likely the source of Wnts for β-catenin activation in hepatocytes for proliferative function.

### 2.3. Wnt–β-Catenin Signaling in Liver Zonation

Another unique characteristic of the liver is the expression of unique genes by the hepatocytes based on their location within a microscopic hepatic lobule. This disparate gene expression allows for the hepatocytes to perform distinct functions that are necessary for the delivery of optimal hepatic output in terms of metabolism, synthesis, and detoxification, which are the broad categories of around 500 functions that hepatocytes perform to maintain health and homeostasis. Toward this end, Wnt–β-catenin signaling is known to be the major regulator of the expression of genes in the zone-3 or pericentral region of the metabolic lobule [26,35,36]. These genes belong to the category of glutamine synthesis, glycolysis, lipogenesis, ketogenesis, bile acid synthesis, heme metabolism, and xenobiotic metabolism. Some of these target genes include Glul, which encodes glutamine synthetase (GS), and is specifically localized to 1–2 layers of hepatocytes around the central vein [37]. To prevent ammonia from leaving the liver, the zone-3 hepatocytes are efficient in its uptake and the high levels of GS in these cells are responsible for condensing ammonia to glutamate, leading to the formation of glutamine. Thus, intracellular levels of glutamine are highest in zone-3 hepatocytes. Some of the other key targets of β-catenin in zone-3 hepatocytes include Axin-2, Lect2, Cyp2e1, Cyp1a2, and others. Recently, choline transporter organic cation transporter 3 was also shown to be a target of the Wnt–β-catenin signaling, which led to the increased uptake of choline by HCC to promote phospholipid formation and DNA hypermethylation, and contributing to hepatocyte proliferation [38]. In fact, several of these β-catenin targets are upregulated in liver tumors where β-catenin signaling is highly activated in both preclinical models and in patients. Conversely, genetic knockout models that lack Wnt secretion from endothelial cells, lack LRP5 and 6 on hepatocytes, or lack β-catenin in hepatocytes, all lack zone-3 targets of the Wnt–β-catenin pathway [23,24,26,27,36]. Wnt2 and Wnt9b appear to be the major drivers of zonated β-catenin activation, and appear to be within the endothelial cells lining the central vein [39].

Thus, broadly, β-catenin seems to be playing a role in hepatocyte proliferation in physiological states including hepatic development (prenatal and postnatal) and liver regeneration (surgical and injury-driven), as well as in regulating hepatocyte functions including basally in the hepatocytes contained in zone-3 of the metabolic lobule. It is pertinent to mention the existence of regulators of the Wnt–β-catenin signaling that have been shown to play a role in the aforementioned hepatic processes. Factors like R-spondins and their receptors LGR4/5 have been shown to potentiate the effects of the Wnt–β-catenin pathways and have been specifically shown to positively impact the processes of both liver regeneration and metabolic zonation [40,41].

## 3. β-Catenin as a Component of the Adherens Junctions in Liver Pathophysiology

In addition to β-catenin being the major effector of Wnt signaling, it plays another evolutionarily conserved role at the adherens junctions (AJs), where it links the cytoplasmic tail of E-cadherin to α-catenin and F-actin [42]. Since the extracellular domain of E-cadherin of one cell binds to its counterpart on the next epithelial cell, the AJs are important mediators of intercellular adhesion. AJs are also present on hepatocytes, which are the predominant functioning epithelial cells of the liver. In fact, β-catenin and E-cadherin are mostly seen at the cell surface of hepatocytes. Immunohistochemistry is rarely sufficiently sensitive to detect β-catenin in cytoplasm or nuclei—even in zone-3 hepatocytes, where it is basally active. β-Catenin clearly associates with E-cadherin in the normal liver, and this association is likely part of maintaining junctional integrity, cell polarity, and epithelial identity, and plays a role in both cell adhesion in addition to providing some barrier function within this highly secretory and vascular organ.

### 3.1. β-Catenin–E-Cadherin Complex in the Liver and Its Regulation

The regulation of β-catenin at the AJs in the hepatocytes is not completely understood. There is an incomplete understanding of whether the same pool of β-catenin is allocated to Wnt signaling and AJs, of when and how this allocation occurs, and of how dynamic this process is [42]. The β-catenin–E-cadherin complex does not seem to be influenced by the Wnt signaling pathway. While liver-specific β-catenin knockout mice showed an absence of β-catenin–E-cadherin interactions, disruption of the Wnt–β-catenin signaling pathway did not impact this complex. This was evident when Wnt co-receptors LRP5/6 were conditionally deleted from hepatocytes, or when Wnt secretion was prevented from hepatic sinusoidal endothelial cells by loss of Wntless [24,27]. In both these models, β-catenin was intact at the AJs and was observed to be interacting with E-cadherin, thus maintaining cell–cell junctions and intact blood–bile barriers. This suggests that the absence of Wnt signaling does not impact the association. Interestingly, tyrosine phosphorylation of β-catenin, especially at tyrosine residue 654, has been shown to play an important role in negatively impacting β-catenin’s association with E-cadherin [43]. Several receptor tyrosine kinases (RTKs), such as Src, EGFR, and Met, have been shown to phosphorylate β-catenin at these residues to negatively impact the AJ assembly, for which β-catenin tyrosine residues 654 and 670 have been shown to be important [44]. The fate of β-catenin following release from this complex is not completely clear, but may function as a co-activator for the TCF family, similar to its role in the Wnt signaling [45]. Indeed, RTKs like HGF and EGF can induce the nuclear translocation and activation of β-catenin signaling to cause liver growth, and can also be seen in a subset of tumors like hepatoblastomas and fibrolamellar HCCs [46,47,48]. Additionally, the cytoplasmic domain of E-cadherin in and around residues 685–699 contains several serine phosphorylation sites, and when these sites are phosphorylated, they interact extensively with armadillo repeats 3–4 of the β-catenin protein [49]. These phosphorylation events may be important in regulating β-catenin–E-cadherin interactions.

### 3.2. γ-Catenin Compensates for β-Catenin at AJs in the Absence of β-Catenin

Another important observation made in the livers of mice lacking β-catenin in hepatocytes was the maintenance of intact AJs. This coincided with an increase in γ-catenin or plakoglobin, a normal inhabitant of the desmosomes. Indeed, in the β-catenin knockouts, γ-catenin was shown to co-precipitate and thus bind to E-cadherin in lieu of β-catenin [50,51]. This was also previously observed in skin and heart [52,53]. To demonstrate the true functionality of the γ-catenin interaction with E-cadherin in the absence of β-catenin, we conditionally knocked out both β- and γ-catenin from liver epithelia using albumin-cre. This led to a severe cholestatic disease, progressive fibrosis, and mortality, which was associated with perturbations in cell–cell junctions, paracellular leaks, and a decrease in E-cadherin [54]. Indeed, β-catenin binds to the region of E-cadherin which contains the PEST sequence motifs, which allow for the recognition of E-cadherin by ubiquitin ligases as well as proteasomal degradation [49]. The binding of β-catenin to E-cadherin masks these motifs and allows for uneventful trafficking of the complex to the AJs. It is likely that γ-catenin binds to the same region of E-cadherin when β-catenin is absent, preventing E-cadherin degradation and successful delivery of the E-cadherin–γ-catenin complex to the cell surface. This also explains the notable decrease in E-cadherin in the β-γ-catenin double-knockout livers [54].

## 4. Hepatocellular Cancer

### 4.1. Alarming Trends in HCC Incidence

The incidence of hepatocellular carcinoma (HCC) has risen steadily in the US and worldwide over last decades [55,56]. Analysis of the NCI’s (National Cancer Institute’s) Surveillance, Epidemiology and End Results (SEER) database reveals alarming trends in HCC incidence. The rates for new liver and intrahepatic bile duct cancer cases have been rising on average 2.7% each year over the last 10 years. Death rates have risen on average 2.6% each year from 2005 to 2014. In 2014, there were an estimated 66,771 people living with liver tumors in USA. In 2020, liver tumors represented 2.4% of all new cancer cases in the US, with around 42,810 new diagnosed cases [55]. Globally, HCC is the 5th most common malignancy in men, 9th most common cause of cancer in women, and the overall 6th most common cancer worldwide [56].

### 4.2. Cellular and Molecular Pathogenesis of HCC

Most HCCs are a consequence of years of hepatic damage and wound healing. The events leading up to HCC are complex and involve bouts of cell injury and death, immune cell infiltration, oxidative stress, and stellate cell activation [57]. The liver tries to replace the dying hepatocytes through chronic regeneration via hepatocyte proliferation. Proliferating hepatocytes are susceptible to DNA damage and mutations, and the associated activation of signaling pathways. Any such alterations that provide proliferative and survival advantage to a cell lead to the initiation of the neoplastic transformation. Transcriptomic and whole-genome sequencing has validated that subsets of HCC are “driven” by key oncogenic signaling pathways [58,59,60]. The whole-exome sequencing of a large number of HCC cohorts has revealed common mutations that are the basis of the molecular classification of HCC [59]. Such analysis has revealed that irrespective of etiology, chronic injury, and downstream cellular events, HCC is driven by a few common genetic aberrations and molecular pathway activation, with only some preferential signaling evident in a few etiologies [60]. One common pathway activated in HCC independent of etiology is Wnt–β-catenin signaling.

## 5. β-Catenin and Hepatocellular Cancer

### 5.1. Mechanism of β-Catenin Activation in HCC

It is important to emphasize the key phosphorylation sites located in exon-3 of β-catenin, which are important in its eventual degradation. When the Wnt signals are absent, β-catenin is sequentially phosphorylated at serine-45 (S45), S33, S37, and threonine-41 (T41) by casein kinase I (CKI) and glycogen synthase kinase 3β (GSK3β) [61]. Phosphorylated β-catenin is recognized by β-transducin repeat-containing protein for ubiquitination and proteasomal degradation, and requires intact D32 and G34 sites [62]. When Wnt signaling is on, it inactivates the β-catenin degradation complex consisting of Axin-1 and adenomatous polyposis coli gene product (APC) in addition to CKI and GSK3β. Around 26–37% of all HCCs display *CTNNB1* mutations [8,63]. These missense mutations are localized to exon-3 of CTNNB1, the gene encoding for β-catenin, and affect phosphorylation and ubiquitination sites in the β-catenin promoter, making it resistant to degradation. This leads to β-catenin stabilization, nuclear translocation, and activation of the downstream target genes, playing important and unique roles in tumor biology in several subsets of HCC cases. There are several targets of β-catenin reported in HCC [8]. Some highly relevant ones include glutamine synthetase (GS), cyclin-D1, VEGF-A, lect2, Axin-2, and others.

Loss-of-function mutations in *AXIN1* are another major contributor to HCC development. *AXIN1* is also among the top five mutated genes in HCC, seen in around 8% of human HCCs. This gene normally encodes for a protein essential for β-catenin degradation. In the absence of a functional Axin-1, β-catenin levels are increased and Wnt signaling is activated. Indeed, in preclinical models which used sleeping beauty transposon/transposase to express shRNA-*Axin1* along with Met proto-oncogene in either a hepatic β-catenin-sufficient or deficient liver, the requirement of β-catenin was unequivocally shown in this model [64]. Intriguingly, only a subset of targets of the β-catenin signaling are positive in these tumors, including cyclin-D1 and c-myc, and interestingly, these tumors are GS-negative.

Analysis of early HCC, multinodular HCC, and comparison of primary and metastatic HCC has also indicated that β-catenin has a trunk role in HCC similar to other major drivers such as mutations in *TERT* promoter or *TP53* [65].

### 5.2. Animal Models to Study β-Catenin Activation in HCC

The hepatic overexpression of β-catenin or the expression of mutated, constitutively-active β-catenin alone is insufficient for HCC development, as reported in many mouse models, suggesting cooperation with other pathways [37,66]. Indeed, *CTNNB1* mutations significantly correlate with the presence of other mutations such as in *TERT* promoter, *NFE2L2, MLL2, ARID2,* and *APOB* [59,67]. *CTNNB1* mutations also are seen to co-occur with the overexpression/activation of Met, Myc, or Nrf2 [63,68,69]. Using a reductionist approach, such concomitant alterations have been modeled in mice by the co-expression of various combinations in vivo using the sleeping beauty transposon/transposase or CRISPR/Cas9 approach and hydrodynamic tail vein injection [70]. For example, 11% of human HCCs show concomitant *CTNNB1* mutations and Met overexpression/activation, and their co-expression in murine liver in the Met–β-catenin model leads to clinically relevant HCC [63,71]. Likewise, Myc–β-catenin represents 6% of human HCCs [68]; and Nrf2–β-catenin represents 9–12% of HCC [69]. The continued generation and characterization of these models for their clinical relevance, biology, and for testing therapies, is of high value.

## 6. State of Therapies for HCC

The five-year survival of liver tumors is 19.6%, attributable partially to lack of effective therapies [55]. For localized disease, partial hepatectomy or liver transplantation are most beneficial. Loco-regional therapies like radio frequency ablation and transarterial chemoembolization are palliative or useful as neoadjuvants. Until recently, sorafenib was the only FDA-approved agent for unresectable HCC, and this non-specific tyrosine kinase inhibitor (TKI) improved survival by 3 months [72]. Several agents have been approved for HCC treatment in the last 5 years. Regorafenib was approved as second-line treatment, showing improvement in survival to 10.6 months vs. 7.8 months for placebo [73]. In 2017, the immune checkpoint inhibitor (ICI) nivolumab was approved by the FDA as second-line treatment, almost doubling overall survival to 15 months in the Checkmate trial [74]. More recently, another TKI, lenvatinib, was approved as first-line therapy, showing non-inferiority to sorafenib [75]. Cabozantinib, a Met inhibitor, also got approval for second-line use in HCC [76]. Another ICI, pembrolizumab, was awarded an accelerated approval as second-line therapy for HCC based on the KEYNOTE-224 trial [77]. More recently, the results of a phase III clinical trial (IMbrave150) showing higher efficacy to sorafenib and a response rate of around 35% to atezolizumab (anti PD-L1) plus bevacizumab (anti-VEGFA) led to their FDA approval as first-line therapy [78]. Some major existing challenges include a lack of biomarker-based therapy to select a proper subset of patients for a specific treatment and to improve response rates to ICIs, which have revolutionized oncology in general.

## 7. Targeting β-Catenin for HCC Treatment

### 7.1. Proof-of-Concept Studies

Because β-catenin is active in a notable subset of HCCs, and is also considered a trunk mutation, its inhibition could have a major impact on the treatment of a subset of these tumors. Several proof-of-concept studies in HCC, both in vitro and in vivo, have demonstrated the relevance of inhibiting β-catenin as a treatment strategy for HCC. siRNA-mediated *CTNNB1* knockdown resulted in a marked decrease in the viability and proliferation of human hepatoma cells in vitro [79]. Similarly, suppressing β-catenin via gamma-guanidine-based peptide nucleic acid antisense also reduced the viability, proliferation, metabolism, and survival of cells of an HCC line [80]. Interestingly, inhibition of β-catenin signaling also resulted in the diminished secretion of angiogenic factors, implying the dual positive effect of such suppression [80]. The DsiRNAs-mediated knockdown of β-catenin mRNA led to a significant decrease of tumor burden in mice bearing ectopic tumors originating from either Hep3B or HepG2 cells [81]. Using a chemical carcinogen (diethylnitrosamine) and tumor promotion (phenobarbital) model which selectively leads to *Ctnnb1*-mutation-driven HCC, β-catenin inhibition using locked nucleic acid antisense (LNA) had a profound impact on tumor development [82]. More recently, using Kras–β-catenin-driven HCC (which highly resembles the Met–β-catenin model), β-catenin was inhibited using EnCore lipid nanoparticles loaded with a Dicer substrate small interfering RNA targeting *CTNNB1*. This led to a notable decrease in tumor burden, also demonstrating β-catenin to be a highly relevant target in HCC for cases driven by *CTNNB1* mutations.

### 7.2. Where to Target Wnt–β-Catenin Signaling in HCC

The most important mechanism of β-catenin activation in HCC are the mutations in *CTNNB1* and the mutations in *AXIN1*. While there have been several other mechanisms identified to modulate β-catenin signaling, including the upregulation of certain Wnt genes, Frizzled genes, and epigenetic loss of negative regulators like DKK and FRPs and others, their true relevance remains unclear since Wnt–β-catenin signaling, like other signaling pathways, is able to regulate its overall activity via robust post-translational mechanisms. However, mutations in *CTNNB1* or *AXIN1* deem the β-catenin protein non-degradable and hence cannot be regulated by the normal mechanisms, which converge on β-catenin degradation to control the signaling pathway activity. This also suggests that several classes of Wnt inhibitors will not work in HCCs because they inhibit or impair Wnt activity upstream of the observed mutations in *CTNNB1* or *AXIN1.* Hence, the suppression of β-catenin itself using the RNA-based therapies discussed in the preceding section, or those impairing β-catenin nuclear translocation, impairing its interaction with TCF4 or preventing the β-catenin–TCF complex from transactivating target genes, would be most effective in treatment of some subsets of HCC. Finally, identifying unique opportunities related to β-catenin signaling in HCC is important, as it may help in selecting or excluding the right group of patients, and may help to identify innovative opportunities to target other mechanisms that are intimately related to β-catenin activation unique to HCC.

### 7.3. How to Target β-Catenin in HCC

Targeting β-catenin itself using RNA-based therapies is highly desirable. Several classes of siRNA- and antisense-based therapies have been described for use against β-catenin. The use of EnCore lipid nanoparticles along with Dicer substrate small interfering RNAs is especially innovative because it can be modified to specifically deliver the payload to liver tumors, and the safety of their use has been shown in patients [83]. Others such as peptide nucleic acid antisense, locked nucleic acid antisense, and other modalities have been reported, and may have eventual clinical use [80,81,82].

There may be an opportunity to identify the mechanisms of the nuclear transport or nuclear export of β-catenin. Targeting molecules that cargo β-catenin to the nucleus or activate its export out of the nucleus could have efficacy in the treatment of β-catenin-mutated HCCs. Pegylated interferon-α2a (peg-IFN), previously a first-line therapy for hepatitis C virus (HCV) patients, was shown to induce the levels of Ran-binding protein 3 (RanBP3), which is known to export β-catenin out of the nucleus [84]. Peg-IFN treatment was also shown to induce association between RanBP3 and β-catenin, and led to decreased TopFlash reporter activity that was abrogated by siRNA-mediated RanBP3 knockdown. In vivo, peg-IFN treatment led to increased nuclear RanBP3, decreased nuclear β-catenin and cyclin D1, and decreased GS, and eventually led to decreased tumor cell proliferation.

The use of small-molecule inhibitors that interfere with its interactions with TCF or other relevant co-factors or components of the transcriptional complex would be highly desirable. However, a high specificity of the small-molecule inhibitors will be required because of the overlap of the β-catenin–TCF4 binding site, and with the binding sites for APC and E-cadherin [85]. Even though a number of the identified compounds showed selectivity of inhibition in vitro (e.g., PKF115-584, CGP049090, and PKF118-310), none of them has entered clinical trials [85]. PR1-724, the next-generation compound of the original small-molecule ICG-001, interferes with β-catenin–TCF4 interactions with CBP, a histone acetyltransferase essential for transcriptional function of the complex [86,87]. PRI-724 has been shown to be safe in patients with HCV-related cirrhosis, and may be of high relevance in the treatment of subsets of HCC with known mutations in *CTNNB1* [88].

### 7.4. Unique and Exploitable Aspects of Targeting β-Catenin in Subsets of HCC

In addition to a general role of β-catenin in regulating tumor cell proliferation, survival, and angiogenesis, there are specific and unique aspects of β-catenin activation due to mutations in HCC which can have notable biological and therapeutic implications—especially related to a step towards precision medicine.

#### 7.4.1. Role of β-Catenin in Tumor Immune Evasion

ICIs have revolutionized the treatment of many tumors, including HCC as can be seen by the FDA approval of nivolumab and pembrolizumab as second-line therapy and of atezolizumab (anti PD-L1) plus bevacizumab (anti-VEGFA), as first line treatment for unresectable HCC [78]. However, there are no available biomarkers which predict either the efficacy or lack thereof to ICIs. Clinical response to ICIs, most of which are T-cell-based therapies, depend on the presence of a CD8^+^ T cell inflamed environment and chemokines and interferon signature within the tumor [89]. Intriguingly, activation of β-catenin signaling has been linked to immune evasion in tumors such as melanoma through T-cell exclusion from tumors [90]. This is shown in our own analysis as well (Figure 1). Several mechanisms underlie this observation, including the effect of β-catenin activation on CD8^+^ T cell priming and infiltration by acting on Batf3-lineage CD103^+^ dendritic cells (DCs) and decreasing CCL4 production by inducing the expression of transcription repressor ATF3 [91]; disruption of Foxp3 transcriptional activity, key for development and function of regulatory T cells [92]; and increased Treg survival, which can reduce CD8^+^ T cell proliferation [93]. HCCs with β-catenin activation have been linked to immune cell exclusion [94,95]. We have shown that *CTNNB1*-mutated HCCs are resistant to anti-PD-1 [68], and hence may benefit from the inhibition of β-catenin or its downstream effectors to sensitize these tumors to ICIs.

One additional relevant mechanism in the liver might be through a known interaction of β-catenin with NF-κB in the hepatocytes and liver tumor cells [96]. This inhibitory association between the p65 subunit of NF-κB and β-catenin prevents NF-κB activation even when appropriate upstream effectors of NF-κB are present. In this study, we also showed that this association led to reduced p65-luciferase reporter activity when constitutively active β-catenin was transfected in hepatoma cells. Furthermore, β-catenin mutated HCCs showed decreased p65 nuclear translocation. Knowing that NF-κB signaling plays a major role in inducing inflammatory milieu [97], its suppression brought about by stable β-catenin due to mutations in *CTNNB1* may be one additional contributor of an immune-deficient tumor microenvironment which may in turn lead to resistance to ICIs.

#### 7.4.2. Role of β-Catenin in Regulating Tumor Metabolism Through mTORC1 in HCC

The suppression of β-catenin in *CTNNB1*-mutant liver tumors decreases tumor burden in many models [71,82]. We made a unique discovery of how this response was mediated by the regulation of mTORC1 by β-catenin [98]. The Wnt–β-catenin pathway transcriptionally regulates the expression of *Glul*, which encodes GS in hepatocytes in zone-3 of the hepatic lobule [37], and leads to the highest glutamine in zone-3 hepatocytes [99] (Figure 2A,B). Glutamine directly phosphorylates mTOR at serine-2448 in lysosomes [100]. We identified p-mTOR-S2448 (active mTORC1) [101] in zone-3 hepatocytes basally, which was absent in hepatocyte-specific knockout (KO) of β-catenin, Wnt co-receptors LRP5-6, and GS (Figure 2B). We also found by immunohistochemistry (IHC) that HCCs with *CTNNB1* mutations are simultaneously positive for GS and p-mTOR-S2448 in preclinical models and patients (Figure 2B). We also showed a dependence of the CTNNB1-mutated HCCs to mTORC1 by their susceptibility to mTOR inhibition by rapamycin in a preclinical model. This may be a novel way to target β-catenin mutated liver tumors in patients until anti-β-catenin therapies become a reality.

## Figures and Tables

**Figure 1 cancers-13-01830-f001:**
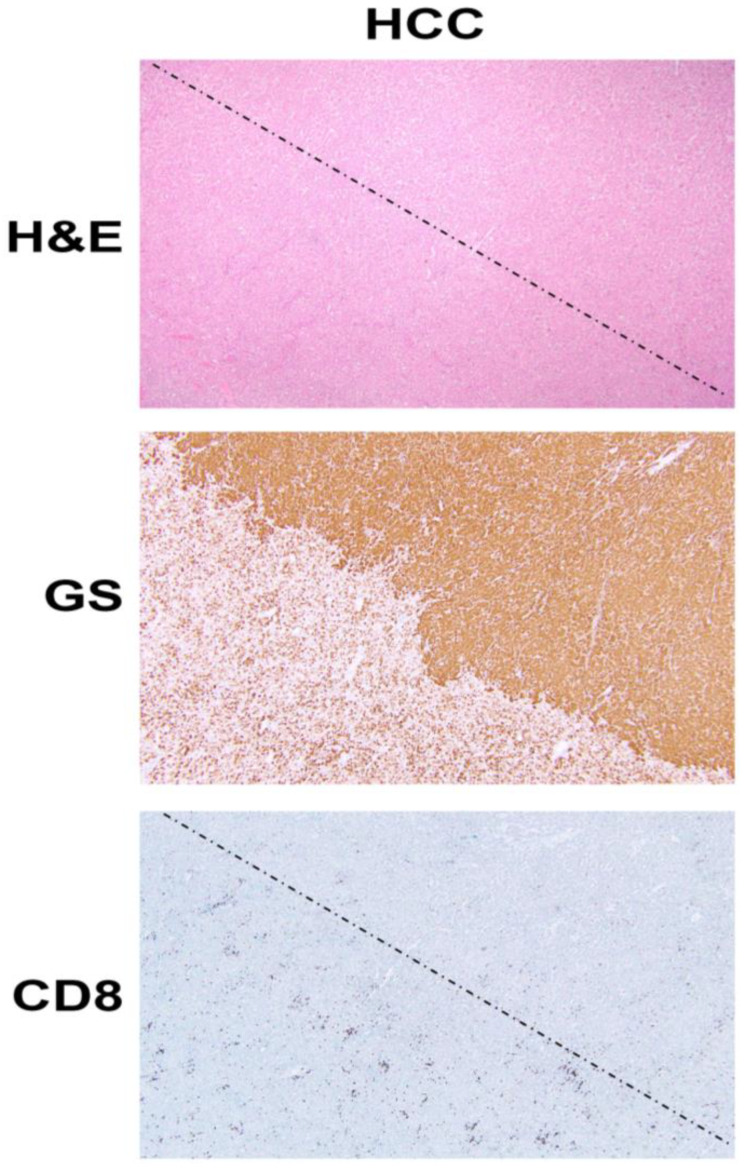
β-Catenin activation in HCC reduces CD8 T cell infiltration in the tumor. The top panel shows histology of explanted liver for hepatocellular cancer (HCC) showing the presence of two distinct tumors (separated by a dotted line) which are otherwise difficult to distinguish and demarcate by hematoxylin and eosin (H&E) staining (100×). The middle panel shows the immunohistochemistry of the adjacent tissue section to the top panel, for glutamine synthetase (GS), a surrogate marker of β-catenin activation due to mutations in *CTNNB1*. The staining for GS shows the presence of uniform positive staining in the upper-right part which decorates a β-catenin-active HCC, whereas the lower-left tumor is negative for this stain. Immunohistochemistry for CD8 for a subset of T cells, in the section adjacent to those shown in the top and middle panels, shows a general dearth of positive cells in the top-right (β-catenin-active) tumor, while there are notably more CD8-positive cells in the lower left or in the non-β-catenin-active HCC. The two tumors are separated by a dotted line.

**Figure 2 cancers-13-01830-f002:**
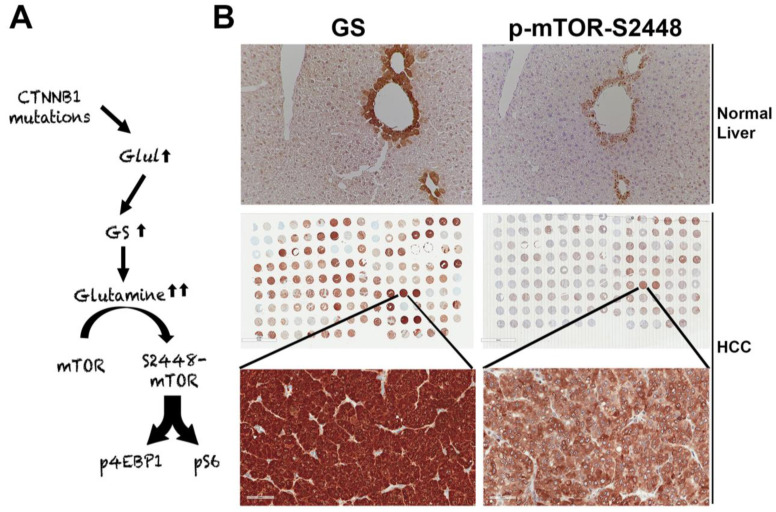
Unique mTORC1 addiction of *CTNNB1*-mutated HCCs due to glutamine. (**A**) The unique axis of mTORC1 activation in β-catenin gene mutated HCCs due to overexpression of *GLUL,* the gene encoding for glutamine synthetase (GS), which generates glutamine from ammonia and glutamate, and in turn glutamine activates mTORC1 in lysosomes. (**B**) The top panel shows immunohistochemistry for GS and p-mTOR-S2448 in adjacent sections from a normal mouse liver. Both proteins are localizing exclusively to zone-3 hepatocytes in the immediate proximity to the central vein (200×). The whole slide scans (middle row) of two adjacent tissue microarrays of human HCC samples stained for the same antibodies against GS and p-mTOR-S2448 also shows several HCCs to be simultaneously positive for GS and p-mTOR-S2448. A representative tissue array sample is magnified (400×) to show GS and p-mTOR-S2448-positive HCC (bottom panels).

## Data Availability

Not applicable.

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
