# Peer review of "β-Catenin Activation in Hepatocellular Cancer: Implications in Biology and Therapy"

_cancers, 2021, doi:10.3390/cancers13081830_

Round 1

Reviewer 1 Report

Hepatocellular cancer (HCC) is an aggressive malignancy with the increasing incidence due to the prevalence of obesity. Wnt signaling pathway plays an important role in the development of HCC. The downstream effector of Wnt signaling, catenin, encoded by CTNNB1, is a very critical player in the Wnt pathway. The detailed study of CTNNB1 has a critical effect on medical practice and is allowed for the development of new therapeutic approaches for the treatment of HCC. Recently, with the rapid development of genomics such as single cell RNA sequencing and GWAS and microRNA research, a growing body of research articles involved in HCC and CTNNB1 have been published, and there is a need to generate a review article to comprehensively describe the progresses in the field of CTNNB1 and HCC. In this article, Dr. Krutsenko et al. provide a comprehensive and updated review on the importance of CTNNB1 in the pathogenesis of HCC as well as the mechanisms of dysregulated CTNNB1. Based on the functional approach, this article meets the journal's scope in general. The manuscript is well written and organized. More importantly, the review article covers the shortage of updated review article of CTNNB1, which will be broadly interesting to researchers this field.

Author Response

We thank the reviewer for their comments. 

Reviewer 2 Report

In this review, Krutsenko et al. analyze the experimental litterature on the impact of beta-catenin activation in the liver and in Hepatocellular cancer. This review adds to this highly-commented field and is well-written by researchers being essential contributors to the field.
The overall organization of the review is fully acceptable. But it is regrettable that the authors took the easy way to favor self-citations, omitting some essential references which should be added.

The references 8-9-10 are previous reviews from the same team. The review from Perugorria in Nat Rev Gastro Hepatol should be added as it is more recent and approach beta-catenin signaling in the liver with another point of view.

To the refs 14-15 which evokes the role of beta-catenin in liver development, that from Decaens, Hepatol 2008 is lacking.

To the point about beta-catenin and liver regeneration, the reference from Torre, J Hepatol 2011 is to be added.

No point is made about the R-Spondin co-ligands of Wnts and about their Lgr4-Lgr5 receptors in the liver, with the essential references of Planas-Paz, Nat Cell Biol 2016 and of Rocha, Cell Reports 2015.

Author Response

Responses below:

  1. But it is regrettable that the authors took the easy way to favor self-citations, omitting some essential references which should be added.

We are sorry that the reviewer feels that way. Where possible, we have included citations as suggested by the reviewer

-The references 8-9-10 are previous reviews from the same team. The review from Perugorria in Nat Rev Gastro Hepatol should be added as it is more recent and approach beta-catenin signaling in the liver with another point of view.

This has been added

-To the refs 14-15 which evokes the role of beta-catenin in liver development, that from Decaens, Hepatol 2008 is lacking.

This has been added

To the point about beta-catenin and liver regeneration, the reference from Torre, J Hepatol 2011 is to be added.

This has been added

No point is made about the R-Spondin co-ligands of Wnts and about their Lgr4-Lgr5 receptors in the liver, with the essential references of Planas-Paz, Nat Cell Biol 2016 and of Rocha, Cell Reports 2015.

This has been added

Reviewer 3 Report

This review by Krutsenko et al. describes the β-Catenin activation in hepatocellular carcinoma. It has potential; the authors are clearly familiar with the subject. However, several issues are needed to be addressed. The topic of the Ms is β-Catenin in cancer, but thorough description was mainly presented for the Wnt/ β-Catenin in normal liver pathophysiology, not in cancer itself.

  1. Unfortunately, some parts of the Ms is weak and unclear. For example, the subchapter of animal models. Authors would need to mention how the liver regeneration and more importantly, the hepatocarcinogenesis, occurred in these mutant mice models. The same problem is noticed for the subchapter “Unique and exploitable aspects of targeting β-catenin in subsets of HCC”. I do not understand the message of it.
  2. What is the effect on β-catenin targeting to the cancer cells? At least in in vitro?
  3. In the end of this review, the authors need to give their conclusion and perspective on the topic. It is very important.

Minor comments:

  1. The MeSH term for HCC is Hepatocellular Carcinoma. I understand that the term Hepatocellular Cancer is also common, so I let the authors decide on the wording.
  2. Please put magnification scale on Figure 1 and 2.
  3. Consistency on the phrasing and abbreviations (e.g. beta-catenin or β-Catenin, HGF, EGF, etc.).
  4. Please pay attention on references (50, 51, and 62).

Author Response

  1. Unfortunately, some parts of the Ms is weak and unclear. For example, the subchapter of animal models. Authors would need to mention how the liver regeneration and more importantly, the hepatocarcinogenesis, occurred in these mutant mice models. The same problem is noticed for the subchapter “Unique and exploitable aspects of targeting β-catenin in subsets of HCC”. I do not understand the message of it.

Response: I am unsure of the first comment. Regarding the second point, the main heading should be "Unique and exploitable aspects of targeting β-catenin in subsets of HCC”. And there are two subheadings of this section:

a. Role of b-catenin in tumor immune evasion.

b.  Role of b-catenin in regulating tumor metabolism through mTORC1 in HCC.

2. What is the effect on β-catenin targeting to the cancer cells? At least in in vitro?

Response: In vitro cell culture studies are really not relevant. Most of the significant work is done in vivo (animal, PDXs,) or in PDOs.

3. In the end of this review, the authors need to give their conclusion and perspective on the topic. It is very important.

I feel this section to be often redundant with abstract. Hence I would prefer not to have this section. However, if the editors will insist, I could write something up.

Minor comments:

  1. The MeSH term for HCC is Hepatocellular Carcinoma. I understand that the term Hepatocellular Cancer is also common, so I let the authors decide on the wording.

No comments

  1. Please put magnification scale on Figure 1 and 2.

Appropriate magnifications are included in figure legend.

  1. Consistency on the phrasing and abbreviations (e.g. beta-catenin or β-Catenin, HGF, EGF, etc.).

We prefer these terms and copy editors can update all of these to β-Catenin, HGF, EGF for sake of consistency

  1. Please pay attention on references (50, 51, and 62).

These have been updated, Thanks.

Round 2

Reviewer 3 Report

The authors had answered my comments. I have no other issues on this manuscript.